# Vaccination impairs de novo immune response to omicron breakthrough infection, a precondition for the original antigenic sin

Jernej Pušnik [1,2] ✉, Jasmin Zorn[1,2], Werner O. Monzon-Posadas[1,2,3], Kathrin Peters [1,2], Emmanuil Osypchuk[1,2], Sabine Blaschke[4] & Hendrik Streeck[1,2]

Several studies have suggested the imprinting of SARS-CoV-2 immunity by original immune challenge without addressing the formation of the de novo response to successive antigen exposures. As this is crucial for the development of the original antigenic sin, we assessed the immune response against the mutated epitopes of omicron SARS-CoV-2 after vaccine breakthrough. Our data demonstrate a robust humoral response in thrice-vaccinated individuals following omicron breakthrough which is a recall of vaccine-induced memory. The humoral and memory B cell responses against the altered regions of the omicron surface proteins are impaired. The T cell responses to mutated epitopes of the omicron spike protein are present due to the high cross-reactivity of vaccine-induced T cells rather than the formation of a de novo response. Our findings, therefore, underpin the speculation that the imprinting of SARS-CoV-2 immunity by vaccination may lead to the development of original antigenic sin if future variants overcome the vaccine-induced immunity.

Less than a year after SARS-CoV-2 emergence first vaccines against the virus were rolled out and administered to individuals at high risk for severe COVID-19[1]. This eventually developed into the largest vaccination campaign in history with over 70% of the global population having received at least one dose of the vaccine by 2023[2]. Vaccines based on novel mRNA technology spearheaded the vaccination campaign demonstrating supreme immunogenicity and protection from severe disease[3–6]. The main component of these vaccines was mRNA encoding the wild-type SARS-CoV-2 spike protein. While vaccination curbed the frequency of severe disease outcomes[7–9] the emergence of SARS-CoV-2 variants, together with waning immune response[10–12], soon rendered vaccine-induced immunity insufficient for protection from infection[13,14]. Currently-circulating omicron SARS-CoV-2 variant and its derivatives led to soaring rates of breakthrough infections due to the high density of mutations in the spike protein facilitating immune escape[15–17]. We and others have previously shown that vaccinated individuals with omicron SARS-CoV-2 breakthrough infection show superior plasma neutralization capacity against the omicron variant[18–21]. It is however unclear whether this is merely due to the boosting of vaccine-induced broadly-specific neutralizing antibodies or also due to the naïve B cell priming and production of antibodies targeting mutated neutralizing epitopes. The question of whether the SARS-CoV-2 immunity can adapt to the mutations found in viral variants rather than remaining locked in an initial clonal repertoire imprinted by the vaccination will prove critical for protection against future SARS-CoV-2 variants. An imprinted immune response could lead to a failure of control over viral replication if a virus mutates to the point where it is still recognized but no longer efficiently neutralized by the adaptive immune system. Such an immunological phenomenon is termed original antigenic sin and is well-described for influenza and

[1]Institute of Virology, University Hospital Bonn, Bonn, Germany. [2]German Center for Infection Research (DZIF), partner site Bonn-Cologne, Braunschweig, Germany. [3]Occupational Medicine Department, University Hospital Bonn, Bonn, Germany. [4]Emergency Department, University Medical Center Goettingen, Goettingen, Germany. ✉e-mail: Jernej.Pusnik@ukbonn.de

dengue virus infection[22]. The first speculations about the imprinting of SARS-CoV-2 immunity by previous infection with seasonal coronaviruses emerged early during the pandemic[23,24] and were later supported with experimental findings[25,26]. With the advent of vaccines and the constant emergence of new variants, the SARS-CoV-2 immunity became increasingly convoluted. Several studies investigated the immune response of previously SARS-CoV-2-infected or vaccinated individuals following infection with an emerging variant and suggested imprinting by the initial antigen exposure[27–32]. Furthermore, the high frequency of breakthrough infections with omicron and its subvariants urged the vaccine makers to adapt their vaccines to omicron and its subvariants, hoping for higher protection from infection. The studies on the bivalent wild-type/omicron mRNA boosters, however, failed to demonstrate increased protection compared to the monovalent wild-type booster vaccination. This was allegedly due to the preferential expansion of wild-type- over omicron-neutralizing antibody titers further pointing in the direction of antigenic imprinting[33–35]. Of note, all of these studies based their conclusions on the increased ratio between wild-type- and omicron-specific antibody titers, merely showing that the pre-existing SARS-CoV-2-specific antibodies unable to bind the omicron variant prevail over the antibodies specifically recognizing mutated regions of omicron proteins. This is not unexpected given the higher frequency of immune challenges with wild-type compared to omicron antigens in those studies and does not demonstrate an impairment of de novo response against the mutated epitopes of omicron proteins. Of note, only in the absence of such a response, a new heavily mutated variant could escape the immune control by the mechanism of the original antigenic sin. Without addressing this critical issue, the current literature remains inconclusive regarding the possible development of original antigenic sin. To investigate whether repeated vaccination with the wild-type-based vaccine imprints the response to the infection with a heavily mutated omicron variant, we assessed the adaptive immune response to omicron breakthrough infection in previously vaccinated individuals and two control groups: vaccinated uninfected and unvaccinated omicron-infected individuals. Utilizing state-of-the-art immunological assays, we directly measured the levels of plasma antibodies, peripheral blood B cells, and T cells specific for the mutated regions of the omicron SARS-CoV-2 surface proteins and compared them between the groups.

## Results

### Vaccination impedes the generation of IgG specific for mutated regions of RBD after omicron breakthrough infection

Vaccination but also previous infection induce a potent antibody response against the spike protein of SARS-CoV-2 which is considered the main mechanism of immune protection against the infection[36–38]. To address the effect of previous SARS-CoV-2 vaccination on the humoral response to omicron breakthrough infection, we measured the level of omicron-RBD-specific (RBD: receptor-binding domain of the spike protein) IgG in the plasma of individuals that had received three mRNA (encoding wild-type spike protein) vaccine doses and subsequently recovered from omicron breakthrough infection (Vacc+O-Inf). We then compared the measurements with two control groups: individuals who received three mRNA (encoding wild-type spike protein) vaccine doses and were not infected with SARS-CoV-2 (Vacc), and individuals who did not get vaccinated, but were infected with omicron (O-Inf) (Fig. 1a). Detailed information on the antigen exposure and sampling time points along with the demographic information is provided in the supplemental table 1. The assessment of omicron-RBD-specific IgG was performed with ELISA and demonstrated significantly lower levels of these antibodies in the O-Inf compared to the other two groups ($p < 0.0001$). No significant difference was observed between the Vacc+O-Inf and Vacc groups (Fig. 1b). Within the O-Inf group 20% of individuals had no detectable omicron-RBD-specific IgG, while all vaccinated individuals had these antibodies above the detection level. The time passed

between the last antigen exposure and sampling was comparable between the groups ensuring that the observed differences were due to different antigen exposure histories (Fig. 1c). Omicron RBD contains 15 mutations altering its surface and consequently antibody recognition[39,40]. We next investigated whether vaccinated individuals adapt their antibody repertoire allowing them to target altered regions of the RBD following omicron breakthrough. To dissect this we developed a competitive ELISA where we first incubated the plasma with increasing concentrations of wild-type RBD and subsequently measured the leftover binding capacity to omicron RBD (Fig. 1d–e). This assay was performed for the samples with detectable omicron-RBD-specific IgG. Our findings demonstrate a higher proportion of IgG binding to omicron but not wild-type RBD (% of total omicron-RBD-specific IgG) in the O-Inf group compared to the Vacc+O-Inf ($p < 0.0001$) and Vacc groups ($p < 0.0001$). No significant differences were observed between the Vacc+O-Inf and Vacc groups. (Fig. 1f). Furthermore, we compared plasma levels of IgG binding omicron but not wild-type RBD in individuals with detectable anti-omicron-RBD IgG. The O-Inf group had significantly higher levels of these antibodies compared to the Vacc+O-Inf ($p < 0.0001$) and Vacc groups ($p < 0.0001$). No significant differences were observed between the latter two groups (Fig. 1g). The proportion of individuals with undetectable IgG binding omicron but not wild-type RBD was 0% for the O-Inf, 53% for the Vacc+O-Inf, and 70% for the Vacc groups. Collectively, these findings indicate higher levels of omicron-RBD-specific IgG among vaccinated individuals with or without omicron breakthrough when compared to only omicron-infected individuals. In contrast to unvaccinated, most of the previously vaccinated individuals did not develop detectable levels of antibodies targeting mutated regions of the omicron RBD following omicron breakthrough infection.

### Vaccination impedes the generation of neutralizing antibodies specific for mutated regions of omicron following breakthrough infection

Neutralization is the main mechanism by which antibodies limit viral replication[41]. We, therefore, assessed the plasma neutralization capacity of the three groups of individuals utilizing a plaque reduction neutralization assay against the omicron SARS-CoV-2. According to the data, the Vacc+O-Inf group had higher plasma neutralization capacity than the O-Inf ($p < 0.0001$) and Vacc ($p < 0.001$) groups. Comparing the latter, the neutralization potency of the Vacc group was higher than that of the O-Inf group ($p < 0.01$) (Fig. 2a). The proportion of individuals with undetectable plasma neutralization capacity was 22% for the O-Inf group, 0% for the Vacc+O-Inf and 2% for the Vacc group. The time passed between the last antigen exposure and sampling was comparable between the groups ensuring that the observed differences were due to different antigen exposure histories (Fig. 2b). We next developed a competitive version of plaque reduction neutralization assay allowing for the measurement of neutralizing antibodies targeting mutated regions of the omicron SARS-CoV-2 surface proteins. We pre-incubated plasma with increasing concentrations of wild-type surface proteins that might contain neutralizing epitopes (spike, membrane, and envelope proteins) and tested its neutralization capacity against the omicron SARS-CoV-2 (Fig. 2c–d). This assay was performed for the samples with detectable plasma neutralization against the omicron SARS-CoV-2. The data show a significantly higher proportion of neutralizing antibodies specific for mutated regions of omicron surface proteins (% of all omicron-neutralizing antibodies) in the O-Inf group compared to the Vacc+O-Inf ($p < 0.0001$) and Vacc ($p < 0.0001$) groups. No significant differences were observed comparing the Vacc+O-Inf and Vacc groups (Fig. 2e). Similar was true for the plasma levels of neutralizing antibodies specific for the mutated regions of omicron surface proteins. The O-Inf group had higher levels of these antibodies compared to the Vacc+O-Inf ($p < 0.001$) and Vacc ($p < 0.0001$) groups. No significant differences were

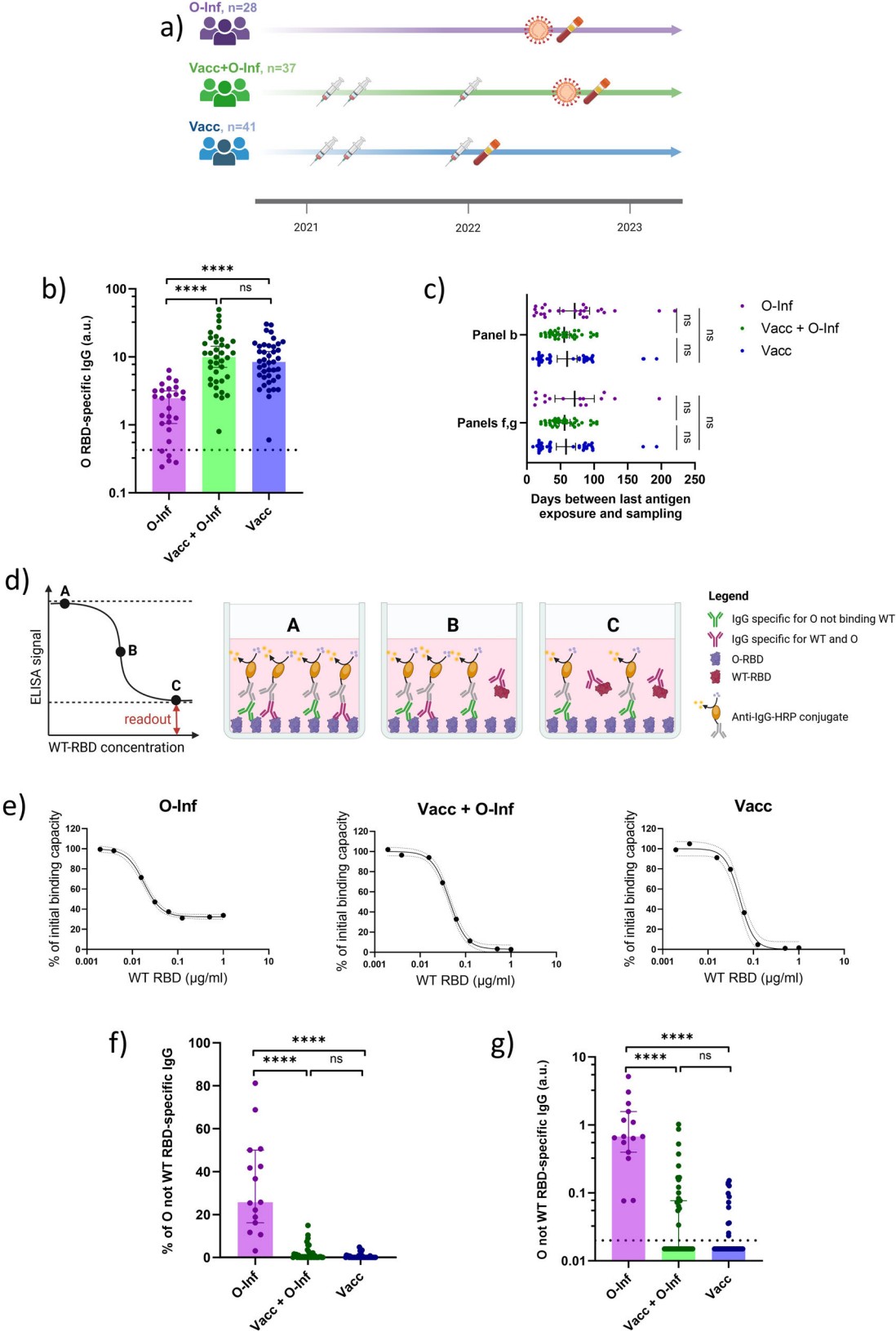

observed comparing the Vacc+O-Inf and Vacc groups (Fig. 2f). Within each group the proportions of individuals with undetectable neutralizing activity against the mutated regions of omicron surface proteins were as follows; 5%, 65%, and 88% for the O-Inf, Vacc+O-Inf and Vacc groups respectively. Taken together these data indicate higher plasma neutralization capacity in vaccinated individuals with

omicron breakthrough infection compared to only vaccinated or infected individuals. Importantly, vaccinated individuals with omicron breakthrough infection have significantly lower levels of antibodies targeting mutated neutralizing epitopes of omicron surface proteins than unvaccinated infected individuals and comparable to vaccinated but uninfected individuals.

**Fig. 1 | Assessment of omicron-SARS-CoV-2-RBD-specific IgG in plasma of three groups of individuals: Inf (infected with omicron, unvaccinated), Inf+Vacc (3-times vaccinated and infected with omicron), Vacc (3-times vaccinated, uninfected).** Omicron SARS-CoV-2 is abbreviated as O and wild-type as WT throughout the figure. **a** Graphical representation of the study design indicating the time points of vaccination (syringe symbol), infection (virus symbol), and sampling (blood vial symbol) for the three groups. **b** Plasma levels of omicron-RBD-specific IgG measured by ELISA for the three groups of individuals. The dashed line represents the positivity cutoff. The following numbers of biologically independent samples were included in each group: n(O-Inf) = 26, n(Vacc+O-Inf) = 37, and n(Vacc) = 41. **c** Time between the last exposure to SARS-CoV-2 antigens, either as vaccination or infection, and sampling time point for individuals included in different panels. The data is displayed as scatter plots with lines indicating mean and 95% confidence intervals. The numbers of biologically independent samples correspond to those used in each panel. **d** Simplified schematic representation of the principle behind the competitive ELISA assay used to determine the proportion of IgG binding omicron but not wild-type RBD. **e** Representative plots demonstrating the percentage of the initial binding capacity of the plasma IgG to omicron RBD following incubation with increasing concentrations of wild-type RBD. A scatter plot with interpolated sigmoidal curve is shown for each group. The 95% confidence intervals around the interpolated curve are displayed as dashed lines **f** Comparison of IgG binding omicron but not wild-type RBD as a percentage of the total omicron-RBD-binding IgG. **g** Plasma titers of IgG binding omicron RBD but not wild-type RBD for the three groups of individuals. The dashed line represents the positivity cutoff. The following numbers of biologically independent samples were included in each group (panels f and g): n(O-Inf) = 15, n(Vacc+O-Inf) = 36, and n(Vacc) = 39. In panels **b**, **f**, and **g** the data is displayed as bar plots indicating the median and 95% confidence intervals with individual data points. Differences between the groups were assessed using the two-sided Kruskal-Wallis test with Dunn's correction for multiple testing. Source data including exact $p$ values are provided as a Source Data file.

## Previous vaccination inhibits the formation of IgG+ B cells specific for mutated regions of omicron RBD following breakthrough infection

Since we observed an impaired humoral response to the mutated regions of the omicron surface proteins in previously vaccinated individuals following omicron breakthrough infection, we next investigated whether this impairment extends to the memory B cell level. We detected IgG+ B cells specific for the wild-type and omicron RBD using multiparameter flow cytometry. Simultaneous staining with both RBD isoforms allowed us to discriminate between the cells specific for only wild-type RBD, only omicron RBD, or both (Fig. 3a). Our findings did not reveal any significant differences in the frequency of omicron-specific IgG+ B cells comparing the three groups (Fig. 3b). Among individuals with detectable IgG+ B cells specific for the omicron RBD, the frequency of cells specific for omicron but not wild-type RBD was decreased in previously vaccinated Vacc+O-Inf ($p < 0.01$) and Vacc ($p < 0.01$) groups when compared to the O-Inf group. No significant difference was observed between the Vacc+O-Inf and Vacc groups (Fig. 3c). The proportion of individuals with undetectable IgG+ B cells specific for omicron but not wild-type RBD was 0% within the O-Inf group, 46% within the Vacc+O-Inf group and 51% within the Vacc group. The time passed between the last antigen exposure and sampling was comparable between the groups ensuring that the observed differences were due to different antigen exposure histories (Fig. 3d). Furthermore, the proportion of IgG+ B cells specific for the mutated regions (% of total omicron-RBD-specific IgG+ B cells) was significantly higher in case of the O-Inf group when compared to Vacc+O-Inf ($p < 0.001$) and Vacc ($p < 0.0001$) groups (Fig. 3e). In summary, IgG+ B cell levels specific for the omicron RBD were similar in vaccinated individuals with or without omicron breakthrough and in unvaccinated omicron-infected individuals. As for humoral immunity, the formation of IgG+ B cells targeting mutated regions of the RBD was inhibited in previously vaccinated individuals.

## The magnitude of T cell response to mutated regions of the omicron spike protein is comparable between the vaccinated individuals with or without omicron breakthrough and unvaccinated omicron-infected individuals

We have demonstrated impaired formation of de novo B cell response to the mutated regions of omicron proteins following omicron vaccine breakthrough infection. T cells comprise another layer of adaptive immune response and were shown to be important for limiting the SARS-CoV-2 infection[42,43]. To investigate whether T cell response to the omicron vaccine breakthrough is also skewed towards the conserved regions of the spike protein we next stimulated PBMC in vitro and measured cytokine expression by flow cytometry. Three different stimuli were used; overlapping peptide pools covering mutated regions of the spike, overlapping peptide pools covering conserved regions of the spike, or negative control without peptides. We differentiated between the CD4 and CD8 T cells. Responsive cells were identified by double expression of IFNg and TNFa (Fig. 4a). Comparing levels of total omicron-spike-specific CD4 T cells, calculated as a sum of responses to conserved and mutated peptide epitopes, we observed a significant increase in the Vacc group when compared to the Vacc+O-Inf group ($p < 0.05$) but not O-Inf group. No difference was observed between the latter two (Fig. 4b). Within each group the proportions of individuals with undetectable CD4 T cell responses against the omicron spike protein were as follows; 0%, 25%, and 12% for the O-Inf, Vacc+O-Inf, and Vacc groups respectively. Among the individuals who successfully mounted omicron-spike-specific CD4 T cell response, we did not observe significant differences in the level of CD4 T cells targeting mutated spike regions between the three groups (Fig. 4c). The time passed between the last antigen exposure and sampling was comparable between the groups ensuring that the observed differences were due to different antigen exposure histories (Fig. 4d). Furthermore, we compared the frequency of CD4 T cells targeting mutated spike regions as proportion of the total CD4 T cell response against the omicron spike protein. The data revealed a significantly higher proportion of these cells in the O-Inf group compared to the Vacc+O-Inf ($p < 0.01$) and Vacc ($p < 0.01$) groups. No differences were observed between the latter two (Fig. 4e). Comparing the levels of total omicron-spike-specific CD8 T cells we observed no significant differences between the three groups (Fig. 4f). Within each group the proportions of individuals with undetectable CD8 T cell response against the omicron spike protein were as follows; 14%, 66%, 54% for the O-Inf, Vacc+O-Inf, and Vacc groups respectively. Among the individuals who successfully mounted omicron-spike-specific CD4 T cell response, no significant differences in the level of CD8 T cells targeting mutated spike regions were observed comparing the three groups (Fig. 4g). The time passed between the last antigen exposure and sampling was comparable between the groups ensuring that the observed differences were due to different antigen exposure histories (Fig. 4h). Also the proportions of CD8 T cells targeting mutated regions (% of the total CD8 T cell response against the omicron spike protein) were comparable between the groups (Fig. 4i). Taken together, our data suggest comparable levels of omicron-spike specific T cells among the vaccinated individuals with or without omicron breakthrough and unvaccinated omicron-infected individuals. In contrast to the B cell response, we did not observe lower frequencies of T cells targeting mutated regions of the omicron spike protein in vaccinated individuals.

## Discussion

The emergence of SARS-CoV-2 variants boosted speculations regarding the role of antigenic imprinting in the pandemic. Drawing parallels from the influenza pandemic studies suggested that immune response

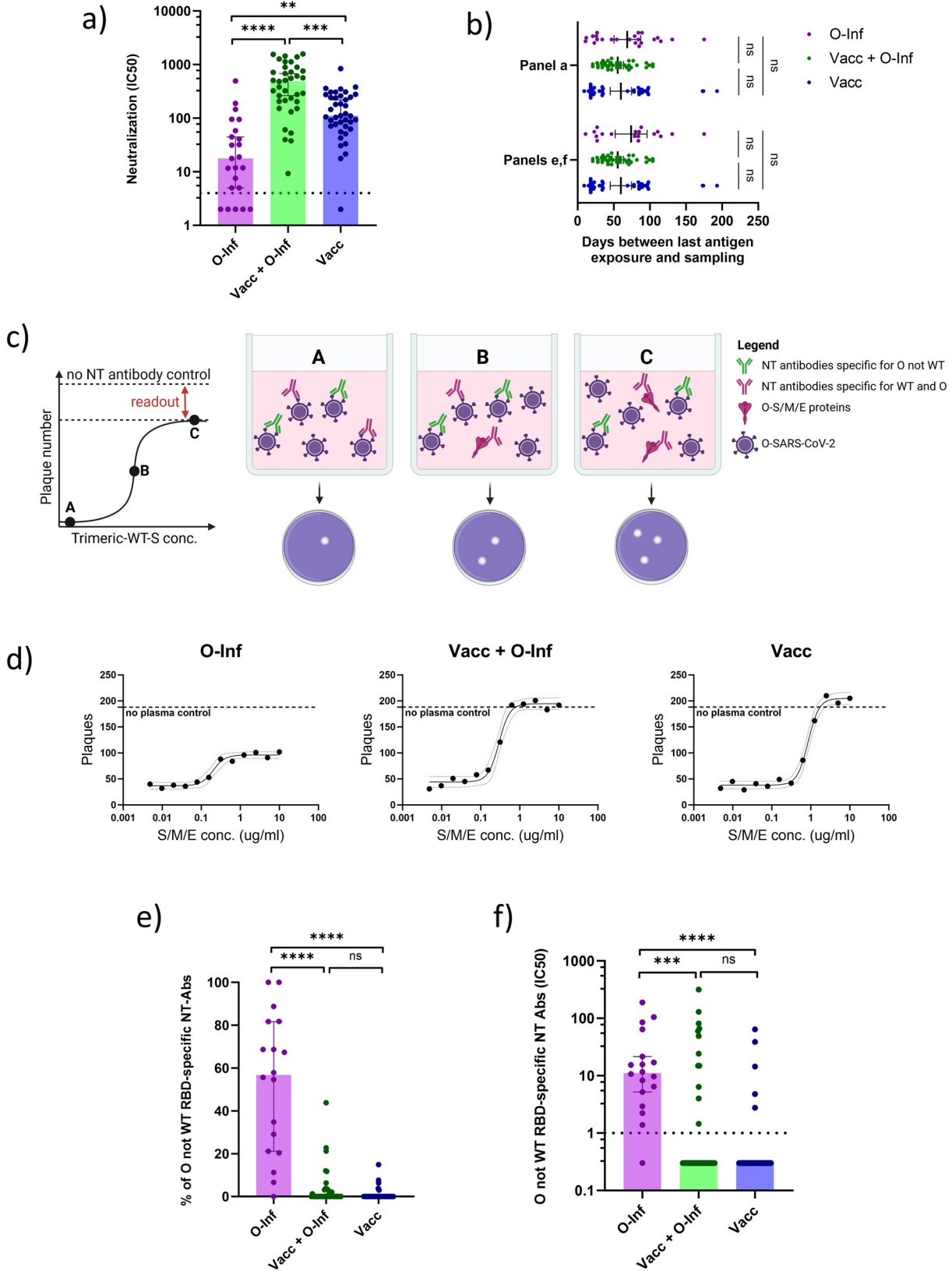

to new variants might be imprinted by exposure to previous viral variants either through infection or vaccination[44,45]. This imprinting may eventually lead to failed control over the replication of a heavily mutated viral variant, an immunological phenomenon known as original antigenic sin. To date, several studies investigating the immune response to breakthrough infection with SARS-CoV-2 variants or booster immunization with variant-adapted vaccines have pointed in the direction of antigenic imprinting by SARS-CoV-2 vaccination by comparing the ratio between wild-type- and omicron-specific antibody titers[27–35]. However, these findings were inconclusive regarding the possible development of original antigenic sin as they did not address other components of the immunological memory, such as memory B

**Fig. 2 | Assessment of omicron-SARS-CoV-2-neutralizing antibodies in the plasma of three groups of individuals: Inf (infected with omicron, unvaccinated), Inf+Vacc (3-times vaccinated and infected with omicron), Vacc (3-times vaccinated, uninfected).** Omicron SARS-CoV-2 is abbreviated as O and wild-type as WT throughout the figure. **a** Plasma neutralization capacity measured by the plaque reduction neutralization assay given as IC50. The dashed line represents the positivity cutoff. The following numbers of biologically independent samples were included in each group: n(O-Inf) = 23, n(Vacc+O-Inf) = 37, and n(Vacc) = 41. **b** Time between the last exposure to SARS-CoV-2 antigens, either as vaccination or infection, and sampling time point for individuals included in different panels. The data is displayed as scatter plots with lines indicating mean and 95% confidence intervals. The numbers of biologically independent samples correspond to those used in each panel. **c** Simplified schematic representation of the principle behind the competitive neutralization assay used to determine the proportion of antibodies that neutralize omicron but do not bind the wild-type SARS-CoV-2 surface proteins; spike (S), membrane (M), and envelope (E). **d** Representative plots demonstrating the number of plaques formed by omicron SARS-CoV-2 following incubation with plasma and increasing concentrations of wild-type surface proteins. A scatter plot with interpolated sigmoidal curve is shown for each group. The 95% confidence intervals around the interpolated curve are displayed as dashed lines. The horizontal dashed line represents the average number of plaques observed in controls without plasma. **e** Comparison of leftover plasma neutralization capacity against the omicron following incubation with wild-type surface proteins as a percentage of initial omicron neutralization. **f** Plasma neutralization capacity against the mutated epitopes of omicron surface proteins. The dashed line represents the positivity cutoff. The following numbers of biologically independent samples were included in each group panels **e** and **f**: n(O-Inf) = 17, n(Vacc+O-Inf) = 37, n(Vacc) = 41. In panels **a**, **e**, and **f**, the data is displayed as bar plots indicating the median and 95% confidence interval with individual data points. Differences between the groups were assessed using the Kruskal-Wallis test with Dunn's correction for multiple testing. Source data including exact *p* values are provided as a Source Data file.

and T cells, and did not adequately address the formation of de novo response against the mutated epitopes of the spike protein. Here, we investigated the adaptive immune response of individuals who had received three doses of mRNA wild-type-based vaccine and were subsequently infected with omicron SARS-CoV-2. We compared their levels of antibodies, B cells, and T cells specific for the mutated regions of the omicron SARS-CoV-2 proteins with those of equally vaccinated uninfected individuals and infected unvaccinated individuals. Our findings demonstrate an impaired B cell response to mutated regions of the omicron SARS-CoV-2 proteins following omicron infection in previously vaccinated individuals. The T cells response to mutated peptides of the spike protein was comparable between the groups indicating higher tolerability of T cells for mutations.

Antibodies targeting the spike protein of SARS-CoV-2 are the best-defined measure of immune protection following infection or vaccination[36–38]. Similar to others we observed higher levels of omicron-RBD-specific IgG in vaccinated individuals with or without omicron SARS-CoV-2 breakthrough infection compared to the only infected individuals underscoring the importance of vaccination[27,31]. In individuals with breakthrough infection, most of the IgG bound the conserved regions of the RBD showing that the antibody response to omicron breakthrough infection is a recall of vaccine-induced memory. Moreover, the level of IgG targeting mutated RBD epitopes was significantly lower than among unvaccinated infected individuals and comparable to the vaccinated uninfected group. This indicates that most vaccinated individuals do not develop humoral responses against the mutated epitopes following omicron breakthrough infection, presumably due to antigenic imprinting. In line with these findings, others have shown unchanged ratios of wild-type/omicron antibody titers following omicron infection in individuals previously vaccinated or infected with wild-type SARS-CoV-2[27–35]. It is noteworthy that some of the vaccinated uninfected individuals had detectable levels of antibodies targeting mutated RBD epitopes although they were never exposed to the omicron SARS-CoV-2. This was presumably due to the presence of preexisting cross-reactive antibodies in plasma that were previously detected in SARS-CoV-2-naïve unvaccinated individuals[46,47]. Overall, we have demonstrated attenuation of de novo humoral response to omicron SARS-CoV-2 breakthrough infection in previously vaccinated individuals.

To investigate whether these findings also apply to neutralizing antibodies we assessed the plasma neutralization capacity against the omicron SARS-CoV-2 for the three groups of individuals. In contrast to RBD-binding antibodies, the plasma neutralization capacity was significantly higher among the individuals with omicron breakthrough infection compared to the control groups. This may be due to the high density of mutations within the omicron RBD and the shift of neutralizing antibody epitopes outside spike RBD[48]. Several previous studies have also demonstrated augmented omicron-neutralizing

potency of plasma following omicron breakthrough infection[18,20,49]. The proportion of neutralizing antibodies targeting mutated regions of the omicron SARS-CoV-2 surface proteins was in line with the RBD-binding antibodies since we did not observe an increase following breakthrough infection. The levels of these antibodies were comparable between the vaccinated groups and lower than in the unvaccinated infected group indicating suppression of de novo neutralizing antibody formation. Some of the vaccinated uninfected individuals had detectable levels of antibodies targeting mutated neutralizing epitopes presumably due to the presence of preexisting cross-reactive antibodies in plasma. Taken together the assessment of omicron-SARS-CoV-2-neutralizing antibodies showed a boosting effect of omicron breakthrough infection and impaired response against the mutated neutralizing epitopes.

We further investigated the effect of antigenic imprinting by SARS-CoV-2 vaccination on B cell response to omicron breakthrough infection. Similar to other studies[35], we did not observe significant differences comparing the frequencies of omicron-RBD-specific IgG+ B cells between the three groups of individuals. Importantly, individuals with omicron breakthrough infection had similar levels of IgG+ B cells binding mutated omicron-RBD regions as uninfected vaccinated individuals and significantly lower than the unvaccinated infected individuals. This further confirmed our observation that the B cell response to omicron breakthrough infection is mostly a recall of vaccine-induced memory and that the response to altered regions of omicron spike protein is inhibited in previously vaccinated individuals.

T cells represent another arm of the adaptive immune response and their role in limiting the SARS-CoV-2 infection is well established[42,43]. We therefore investigated the effect of previous vaccination on omicron-spike-specific CD4 and CD8 T cell responses following omicron SARS-CoV-2 breakthrough infection. In concordance with previous publications, we did not observe the boosting of omicron-spike-specific T cell frequencies by subsequent immune challenges[18,27,31,35]. The frequency of both T cell subsets specific for the mutated omicron spike epitopes among the vaccinated individuals with omicron breakthrough infection was, surprisingly, comparable to those of both control groups. This is most likely due to the lower specificity of the T cells compared to B cells and consequently higher tolerance to mutations within the epitopes. In line with this, a high degree of cross-reactivity has been previously observed for SARS-CoV-2-specific T cells[43,50,51]. In the case of CD4 T cells the proportion of omicron-spike-specific cells that target mutated epitopes was higher in unvaccinated infected individuals indicating that CD4 T cell response is mostly a recall of vaccine-induced T cell memory. To sum up, we observed T cell response to mutated regions of the omicron spike in vaccinated individuals following omicron breakthrough infection. However, this was likely due to the higher tolerability of T cells for

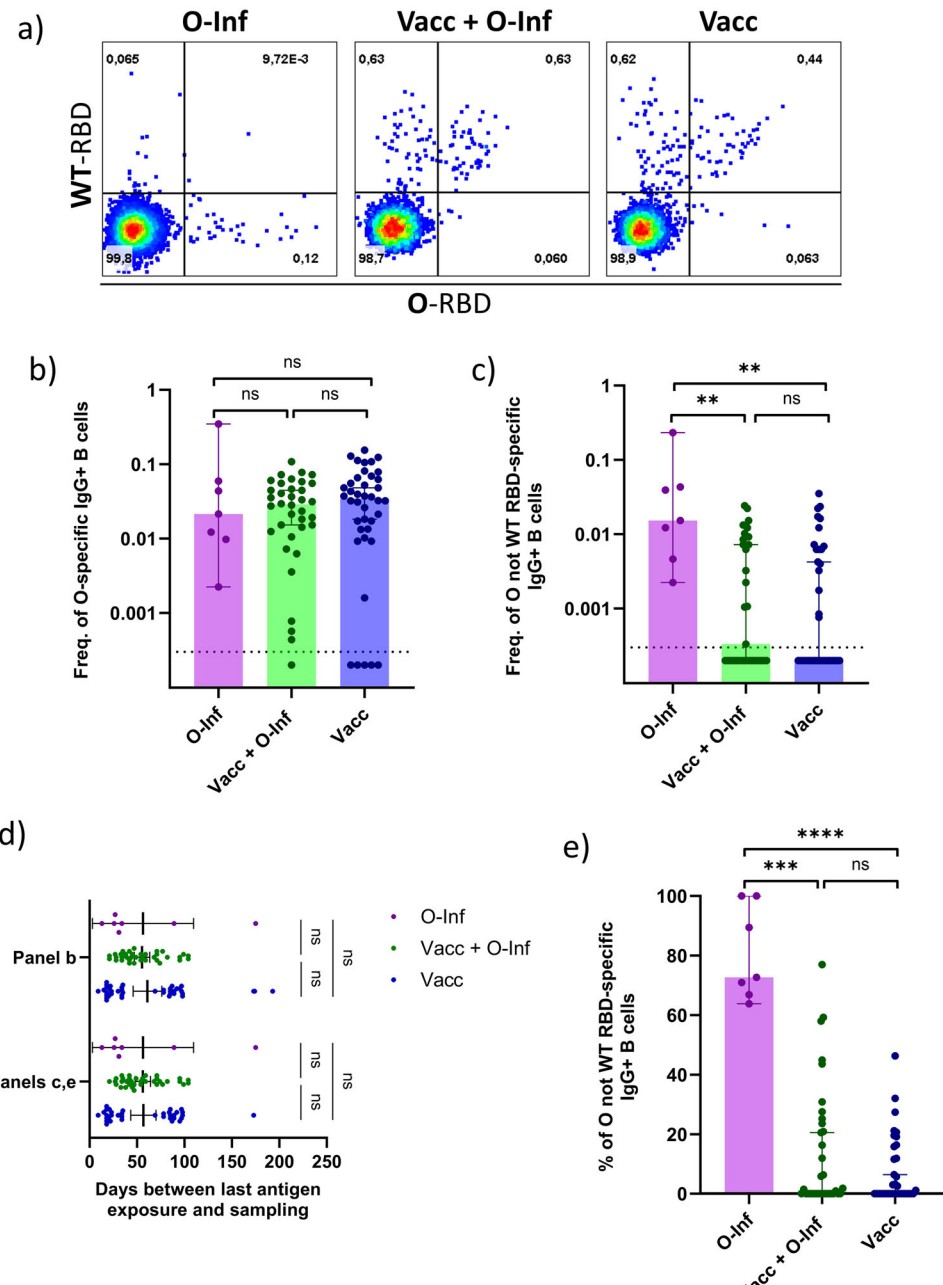

**Fig. 3 | Omicron- and wild-type-RBD-specific IgG + B cells in peripheral blood of three groups of individuals: Inf (infected with omicron, unvaccinated), Inf +Vacc (3-times vaccinated and infected with omicron), Vacc (3-times vaccinated, uninfected).** Omicron SARS-CoV-2 is abbreviated as O and wild-type as WT throughout the figure. **a** Representative flow cytometry pseudocolor plots demonstrating the detection of IgG+ B cells specific for omicron and/or wild-type RBD. Percentages of the parent population are indicated within the gates. **b** Omicron-RBD-specific IgG+ B cells as a percentage of all B cells. The following numbers of biologically independent samples were included in each group: n(O-Inf) = 7, n(Vacc+O-Inf) = 36, n(Vacc) = 40. **c** Omicron-not-wild-type-RBD-specific IgG + B cells as a percentage of all B cells. The following numbers of biologically independent samples were included in each group: n(O-Inf) = 7, n(Vacc+O-Inf) = 34, n(Vacc) = 35. The dashed lines represent the positivity cutoff. **d** Time between the

last exposure to SARS-CoV-2 antigens, either as vaccination or infection, and sampling time point for individuals included in different panels. The data is displayed as scatter plots with lines indicating mean and 95% confidence intervals. The numbers of biologically independent samples correspond to those used in each panel. **e** IgG+ B cells binding omicron but not wild-type RBD as a percentage of all omicron-RBD-specific IgG+ B cells. The following numbers of biologically independent samples were included in each group: n(O-Inf) = 7, n(Vacc+O-Inf) = 34, n(Vacc) = 35. In panels **b**, **c**, and **e** the data is displayed as bar plots indicating the median and 95% confidence interval with individual data points. Differences between the groups were assessed using the Kruskal-Wallis test with Dunn's correction for multiple testing. Source data including exact *p* values are provided as a Source Data file.

mutations rather than the de novo response to mutated spike regions. This high tolerability of T cells for mutations might prove crucial for the maintenance of immunity against future SARS-CoV-2 variants especially in case of a fully developed original antigenic sin within the B cell compartment.

In the present study, we investigated the effect of previous vaccination on the adaptive immune response to omicron SARS-CoV-2 breakthrough infection. Our findings demonstrate that previous vaccination leads to higher titers of neutralizing antibodies, which might reduce susceptibility of these individuals to further SARS-CoV-2

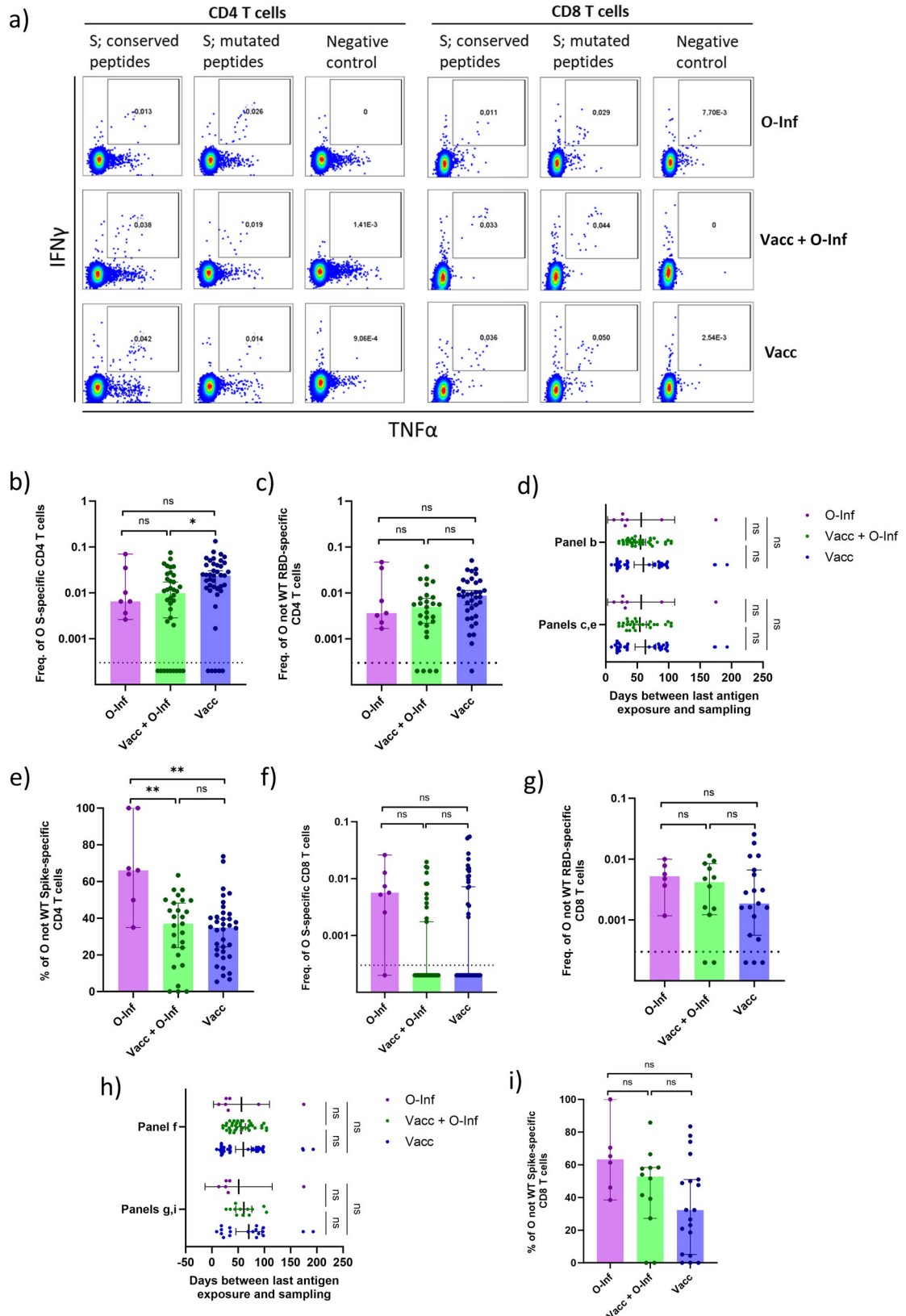

infections. On the other hand, we have shown that vaccination imprints the B cell response to omicron SARS-CoV-2 breakthrough infection and hinders the production of antibodies and memory B cells specific for the mutated epitopes of the omicron surface proteins. This could lead to the development of the original antigenic sin in case the virus mutates to the point where it will no longer be efficiently neutralized by broadly specific antibodies. In contrast to B cells, we observed T cell response to mutated epitopes of the omicron spike protein. However, this was due to the high cross-reactivity of T cells rather than the formation of de novo response further supporting imprinting of the adaptive immunity. In summary, these data show that the imprinting of SARS-CoV-2 immunity by vaccination could lead to

**Fig. 4 | Omicron spike(S)-specific T cells in peripheral blood of three groups of individuals: Inf (infected with omicron, unvaccinated), Inf+Vacc (3-times vaccinated and infected with omicron), Vacc (3-times vaccinated, uninfected).**
Omicron SARS-CoV-2 is abbreviated as O and wild-type as WT throughout the figure. **a** Representative flow cytometry pseudocolor plots demonstrating the detection of T cells specific for the omicron spike (S) protein. Percentages of the parent populations are indicated within the gates. **b** Frequency of omicron-spike-specific CD4 T cells as a percentage of all T cells. The following numbers of biologically independent samples were included in each group: n(O-Inf) = 7, n(Vacc+O-Inf) = 36, n(Vacc) = 41. Frequency of omicron-not-wild-type-spike-specific CD4 T cells as (**c**) percentage of all T cells, (**e**) percentage of all omicron-spike-specific CD4 T cells. The following numbers of biologically independent samples were included in each group: n(O-Inf) = 7, n(Vacc+O-Inf) = 26, n(Vacc) = 36. **d** and **h** Time between the last exposure to SARS-CoV-2 antigens, either as vaccination or infection, and sampling time point for individuals included in different panels. The data is displayed as scatter plots with lines indicating mean and 95% confidence intervals. The numbers of biologically independent samples correspond to those used in each panel. **f** Omicron-spike-specific CD8 T cells as percentages of all T cells. The following numbers of biologically independent samples were included in each group: n(O-Inf) = 7, n(Vacc+O-Inf) = 35, n(Vacc) = 41. Frequency of omicron-not-wild-type-spike-specific CD8 T cells as (**g**) percentage of all T cells, (**i**) percentage of all omicron-spike-specific CD8 T cells. The following numbers of biologically independent samples were included in each group: n(O-Inf) = 6, n(Vacc+O-Inf) = 12, n(Vacc) = 19. In panels **b**, **c**, **e**, **f**, **g**, and **i** the data is displayed as bar plots indicating the median and 95% confidence interval with individual data points. Differences between the groups were assessed using the Kruskal-Wallis test with Dunn's correction for multiple testing. Source data including exact *p* values are provided as a Source Data file.

---

the development of original antigenic sin if future variants overcome the vaccine-induced immunity. As this would inhibit the formation of adaptive immune response, infections with escape variants could become life-threatening. Therefore, our findings call for the development of variant-adapted vaccines that would not only boost the response to conserved regions of the spike protein but also evoke a de novo response targeting mutated epitopes as previously proposed by others[52,53]. Using variant-adapted RBD instead of the entire spike protein as the main vaccine component might overcome the OAS and induce a response towards mutated epitopes since our data suggest that the epitopes of omicron-neutralizing antibodies shift outside the heavily mutated RBD in individuals with omicron breakthrough infection. Another alternative or complementary approach would be to mutate the conserved regions of the spike protein thus preventing the recall of immunological memory and inhibition of de novo response. One of the limitations of this study is the low sample numbers, particularly in the O-Inf group. This was due to the rarity of individuals who recovered from omicron infection without being previously exposed to SARS-CoV-2 antigens. Furthermore, the range of time passed between the last antigen exposure and sampling was wide for some groups which could contribute to the lack of significance. Nevertheless, our study provides a crucial advance in studying the imprinting of SARS-CoV-2 immunity by not only assessing the antibody but also B cell and T cell responses, using competitive ELISA and neutralization assays to directly detect responses directed towards the mutated epitopes and comparing the findings to control groups of either only infected or only vaccinated individuals.

## Methods

### Study cohort
A total of 106 individuals were included in this study. 87 were recruited by the occupational healthcare department of the University Hospital Bonn and 19 by the Emergency Medicine department of the University Göttingen in Germany. The first contact was established by telephone after which a written invitation and a consent form were sent to each participant. Individuals were divided into three groups according to their histories of exposure to SARS-CoV-2 antigens: individuals who had received three mRNA (encoding wild-type spike protein) vaccine doses and subsequently recovered from an omicron breakthrough infection (Vacc+O-Inf, n = 37), individuals who received three mRNA (encoding wild-type spike protein) vaccine doses and were not infected with SARS-CoV-2 (Vacc, n = 41), and individuals that did not get vaccinated but were infected with omicron (O-Inf, n = 28). Age or sex was not among the selection criteria. Following gender distribution was observed between the groups: 65% females and 35% males for the Vacc+O-Inf group, 66% females and 34% males for the Vacc group, 50% females and 50% males for the O-Inf group, and 57% females and 43% males for the subgroup of 7 O-Inf individuals with available PBMC samples. No significant differences in age distribution were observed between the groups (mean years±SD for O-Inf, Vacc+O-Inf, Vacc groups and a subgroup of 7 O-Inf individuals with available PBMC samples respectively: 50 ± 21, 40 ± 15, 47 ± 14, 44 ± 13). SARS-CoV-2 infections were confirmed by RT-PCR. During the period of sample collection, the prevalence of omicron variants was >99% as assessed by sentinel sequencing. Detailed information on the vaccination, infection, and sampling time points as well as demographic information is provided in supplemental table 1. All individuals with omicron SARS-CoV-2 infection did not have previously confirmed SARS-CoV-2 infection. For the Vacc group only individuals without confirmed SARS-CoV-2 infection, and negative nucleocapsid ELISA results were included. Vaccinations of individuals included in this study were performed at the occupational healthcare department of the University Hospital Bonn.

### Ethics approval
The study was approved by the Ethics Committee of the Medical Faculty of the University of Bonn (ethics approval number 125/21) and the Ethics Committee of University Medical Center Goettingen (ethics approval number 21/06/22). All participants provided written informed consent. No compensation was provided for the participants.

### Sample collection and storage
Study participants provided peripheral blood specimens that were centrifuged for 10 min at 600 g to collect plasma. EDTA plasma was stored until analysis at −80 °C. PBMC were isolated by density gradient centrifugation using SepMate™ (Stemcell, 85450) tubes with density gradient medium (Pancoll, PAN-Biotech, P04-60500). The blood was diluted with PBS containing 2% FCS, carefully layered on top of the density gradient medium, and centrifuged at 1200 g for 10 min. The top layer containing the PBMCs was poured off and washed twice with PBS containing 2% FCS. Washed PBMC were resuspended in FCS containing 10% DMSO and frozen at −80 °C overnight. For long-term storage, PBMC samples were transferred to liquid nitrogen.

### Assessment of omicron-SARS-CoV-2-RBD-specific IgG in plasma
An in-house quantitative ELISA was used for the determination of omicron-SARS-CoV-2-RBD-specific IgG. First, microtiter plates with high binding capacity were coated with 100 μl of coating buffer (carbonate-bicarbonate buffer, pH=9.6) containing 1 μg/ml of recombinant omicron SARS-CoV-2 RBD protein (SARS-CoV-2 Spike RBD, His Tag (B.1.1.529/Omicron), Acro Biosystems, SPD-C82E8) and incubated overnight at 4 °C. After washing with wash buffer (PBS with 0.05% (v/v) Tween®-20) plates were blocked (PBS containing 1% (w/v) BSA) to prevent unspecific binding. Cryopreserved EDTA plasma samples were thawed and diluted 400-fold in the blocking buffer. After blocking, plates were washed, incubated with plasma samples, standard dilutions, and negative control (Human IgG Isotype Control, Invitrogen, 12-000-C, 100 ng/ml), washed again, and incubated with 100 μl of HRP-conjugated anti-IgG antibody (Goat anti-Human IgG (Heavy chain) Secondary Antibody, HRP, Invitrogen, A18805) diluted 8000-fold in wash buffer. All incubation steps were

performed at 37 °C for 1 hour. Finally, plates were washed and 100 μl of the substrate solution was added (TMB Chromogen Solution, Life technologies, 002023). The substrate conversion took place at room temperature for 5 min until the addition of 50 μl of 0.2 M H$_2$SO$_4$. The optical density at 450 nm (OD$_{450}$) was measured using Synergy 2 Multimode Plate Reader (BioTek). The background-subtracted OD$_{450}$ readings were interpolated to the standard dilution curve. The positivity cutoff was determined as the mean plus two standard deviations of plasma samples from healthy individuals collected before the COVID-19 outbreak.

### Measurement of plasma IgG specific for the omicron but not wild-type SARS-CoV-2 RBD

An in-house competitive ELISA was used for the determination of IgG specific for the omicron but not wild-type SARS-CoV-2 RBD. Microtiter plates with high binding capacity were coated with 100 μl of coating buffer (carbonate-bicarbonate buffer, pH=9.6) containing 1 μg/ml of recombinant omicron SARS-CoV-2 RBD protein (SARS-CoV-2 Spike RBD, His Tag (B.1.1.529/Omicron), Acro Biosystems, SPD-C82E8) or 1 μg/ml of BSA and incubated overnight at 4 °C. After washing with wash buffer (PBS with 0.05% (v/v) Tween®-20) plates were blocked (PBS containing 1% (w/v) BSA) to prevent unspecific binding. Cryo-preserved EDTA plasma samples were thawed and diluted in the blocking buffer. The plasma dilutions were calculated based on the previous measurement of omicron-SARS-CoV-2-RBD-specific IgG to achieve the OD$_{450}$ of 2. Diluted plasma was then incubated with serial dilutions of wild-type RBD protein (SARS-CoV-2 (COVID-19) S protein RBD, His Tag, Acro Biosystems, SPD-C52H1). A total of 8 dilutions between 1 μg/ml and 0,002 μg/ml were measured for each sample. No further technical replicates were performed. Blocked RBD-coated plates were washed, incubated with plasma samples, standard dilutions and negative control (Human IgG Isotype Control, Invitrogen, 12-000-C, 100 ng/ml), washed again, and incubated with 100 μl of HRP-conjugated anti-IgG antibody (Goat anti-Human IgG (Heavy chain) Secondary Antibody, HRP, Invitrogen, A18805) diluted 8000-fold in wash buffer. BSA-coated plates were incubated with three replicates of diluted plasma samples without wild-type RBD and treated equally. All incubation steps were performed at 37 °C for 1 hour. Finally, plates were washed and 100 μl of the substrate solution was added (TMB Chromogen Solution, Life technologies, 002023). The substrate conversion took place at room temperature for 5 min until the addition of 50 μl of 0.2 M H$_2$SO$_4$. The optical density at 450 nm was measured using Synergy 2 Multimode Plate Reader (BioTek). The background-subtracted OD$_{450}$ readings were interpolated to the standard dilution curve. For each plasma sample incubated with wild-type RBD dilution series a scatter plot was generated and a sigmoidal curve was fitted to determine the top (representing the signal from total omicron-RBD-specific IgG) and bottom (representing the signal from omicron-not-wild-type-RBD-specific IgG) plateaus of the curve. GraphPad Prism software version 9.4.1. (681) was used for this purpose. The background signal of the BSA control was then subtracted from the bottom and top plateaus after which the two values were divided to obtain the proportion of omicron-not-wild-type-RBD-specific IgG relative to the total omicron-RBD-specific IgG. This fraction was multiplied with the corresponding quantitative ELISA measurement to obtain the level of omicron-not-wild-type-RBD-specific IgG in plasma.

### Assessment of omicron-SARS-CoV-2-neutralizing antibodies in plasma

The plasma neutralization capacity was determined by a plaque reduction neutralization assay. Therefore, plasma was heat-inactivated for 30 min at 56 °C and serially two-fold diluted in OptiPRO SFM (Gibco, 12309-019) cell culture medium. A total of 10 dilutions between 4-fold and 32768-fold were measured for each sample depending on the neutralization capacity of a specimen. No further technical

replicates were performed. Each plasma dilution was combined with 80 plaque-forming units of omicron SARS-CoV-2 (B.1.1.529 in OptiPRO SFM (Gibco, 12309-019) serum-free cell culture medium, incubated for 1 h at 37 °C, and added to Vero E6 cells (ATCC, CRL-1586). The cells were seeded in 24-well plates at a density of 1.25×10$^5$ cells/well 24 h earlier. Following 1 h incubation at 37 °C, the inoculum was removed and cells were overlaid with a 1:1 mixture of 1.5% (w/v) carboxymethylcellulose in 2xMEM supplemented with 4% FCS. After incubation at 37 °C for four days, the overlay was removed and the cells were fixed using a 6% formaldehyde solution. Fixed cells were stained with 1% crystal violet solution revealing the formation of plaques. The number of plaques was plotted against the plasma dilutions, and the half-maximal inhibitory concentration (IC50) was determined using GraphPad Prism software version 9.4.1. (681).

### Measurement of neutralizing antibodies specific for the omicron but not wild-type SARS-CoV-2 in plasma

To measure the proportion of neutralizing antibodies that recognize mutated regions of the omicron SARS-CoV-2 surface proteins we developed a competitive plaque reduction neutralization assay. First, plasma was heat-inactivated for 30 min at 56 °C and diluted in OptiPRO SFM (Gibco, 12309-019) serum-free cell culture medium. The plasma dilutions were calculated based on the previous measurement of plasma neutralization capacity against the omicron-SARS-CoV-2 to achieve the 80% neutralization effect. Diluted plasma was then incubated with 12 serial 2-fold dilutions of wild-type SARS-CoV-2 surface proteins, spike (Acro Biosystems, SPN-C52H7), membrane (RayBiotech, YP_009724393) and envelope (Acro Biosystems, ENN-C5128) starting with 10ug/ml and incubated overnight at 4 °C. Plasma sample dilutions, standard dilutions, and negative controls (media without plasma) were combined with 80 plaque-forming units of omicron SARS-CoV-2 (B.1.1.529) in OptiPRO SFM (Gibco, 12309-019) serum-free cell culture medium, incubated for 1 h at 37 °C, and added to Vero E6 cells (ATCC, CRL-1586). The cells were seeded in 24-well plates at a density of 1.25×10$^5$ cells/well 24 h earlier. Following 1 h incubation at 37 °C, the inoculum was removed and cells were overlaid with a 1:1 mixture of 1.5% (w/v) carboxymethylcellulose in 2xMEM supplemented with 4% FCS. After incubation at 37 °C for four days, the overlay was removed and the cells were fixed using a 6% formaldehyde solution. Fixed cells were stained with 1% crystal violet solution revealing the formation of plaques. The number of plaques was plotted against the concentration of the surface proteins and a sigmoidal curve was interpolated using GraphPad Prism software version 9.4.1. (681). The top (representing the signal from omicron-not-wild-type-neutralizing antibodies) and bottom (representing the signal from total omicron-neutralizing antibodies) plateaus of each curve were interpolated from a standard curve and divided to obtain the proportion of omicron-not-wild-type-neutralizing antibodies relative to the total omicron-neutralizing antibodies. This fraction was then multiplied with the corresponding quantitative IC50 to obtain the level of omicron-not-wild-type-neutralizing antibodies in plasma.

### B cell isolation

Cryopreserved PBMC samples were thawed and rested overnight at 37 °C. The next morning, B cells were isolated immunomagnetically (REAlease® CD19 MicroBead Kit, human, Miltenyi Biotec, 130-117-034) following the manufacturer's instructions. Briefly, cells were resuspended in the recommended isolation buffer, labeled with anti-CD19 antibodies coupled to magnetic beads, and passed through a magnetic column. B cell-depleted flow-through was collected for the assessment of T cell responses. Immobilized B cells were washed out of the column and enzymatically released from the beads.

### Detection of SARS-CoV-2 RBD-specific B cells by flow cytometry

To detect the IgG+ B cells specific for the omicron and wild-type SARS-CoV-2 RBD the magnetically isolated B cells were resuspended

in FACS buffer (PBS supplemented with 2% FCS, 0.05% NaN$_3$, and 2 mM EDTA) and incubated with the fluorescently labeled recombinant RBD proteins (Biotinylated SARS-CoV-2 Spike RBD Protein, Acrobiolabs, SPD-C82E8 and Biotinylated SARS-CoV-2 Spike RBD (B.1.1.529/Omicron), Acrobiolabs, SPD-C82E4). The wild-type RBD protein was conjugated with streptavidin-PE (Biolegend, 405204) and omicron RBD with streptavidin-APC (Biolegend, 40520). 15 min into incubation with RBD proteins, an anti-IgG-BV421 antibody (clone G18-145, Biolegend, 562581, diluted 1:20) was added and the incubation was continued for another 15 min. Cells were then washed with PBS and stained for viability (ZombieAqua, Biolegend, 423102) for 15 min at 4 °C. Afterward, cells were washed with FACS buffer and incubated with a solution of antibodies blocking human Fc receptors (FcR block, Miltenyi Biotec, 130-059-901, diluted 1:10) for 10 min at 4 °C. Next, a mixture of fluorescently labeled antibodies consisting of: anti-CD3-BV510 (clone UCHT1, Biolegend, 300448, diluted 1:40), anti-CD27-BV605 (clone O323, Biolegend, 302830, diluted 1:20), anti-IgM-BV785 (clone MHM-88, Biolegend, 314544, diluted 1:20), anti-IgA-VioBright 515 (clone REA1014, Miltenyi Biotec, 130-116-886, diluted 1:40), anti-CD21-PE-Cy7 (clone Bu32, Biolegend, 354912, diluted 1:160), and anti-CD19-APC-Cy7 (clone HIB19, Biolegend, 302218, diluted 1:80) was added. Each antibody was checked for performance and titrated before use. Following incubation at 4 °C for 15 min, the cells were washed again and acquired on a BD FACS Celesta flow cytometer with BD FACSDiva™ Software Version 8.0 (BD Bioscience). Possible longitudinal fluctuations in laser intensity were monitored daily before the experiment using fluorescent beads (Rainbow beads, Biolegend, 422905). The data were analyzed with the FlowJo Software version 10.0.7 (TreeStar). To compensate for the background binding of IgG+ B cells to the fluorescent probes 16 samples were stained with unconjugated streptavidin-PE/APC. The average frequency of streptavidin-PE/APC-binding cells plus two standard deviations was subtracted from the frequencies of RBD-binding cells. No technical replicates were performed due to the scarcity of the samples.

### Ex vivo stimulation of T cells

B-cell-depleted PBMC fraction was seeded in 96-well U bottom plates and stimulated with two different pools of overlapping peptides: the first covering the mutated regions of the omicron SARS-CoV-2 spike protein (PepTivator® SARS-CoV-2 Prot_S B.1.1.529/BA.1 Mutation Pool, Miltenyi Biotec, 130-129-928) and the second covering conserved regions of the spike (PepTivator® SARS-CoV-2 Prot_S B.1.1.529/BA.1 WT Reference Pool, Miltenyi Biotec, 130-129-927). One million cells were stimulated per condition. The final concentration of each peptide was 1 μg/ml for both peptide pools. Co-stimulatory antibodies (BD FastImmune™ CD28/CD49d, BD Bioscience, 347690) were added to a final concentration of 1 μg/ml. For each sample, an equally treated DMSO-stimulated negative control was included. As a positive control, cells were stimulated with PMA (20 ng/ml) (Sigma-Aldrich, P1585-1MG) and ionomycin (1 μg/ml) (Sigma-Aldrich, I3909-1ML). Stimulation was performed at 37 °C for 6 hours. One hour into stimulation Golgi Stop (BD Bioscience, 554724) and Golgi Plug (BD Bioscience, 555029) were added (final concentration 1 μg/ml) to inhibit vesicular transport and prevent the secretion of the cytokines from cells.

### Detection of SARS-CoV-2-specific T cells by flow cytometry

Stimulated cells were washed with PBS and stained for viability (ZombieAqua, Biolegend, 423102) for 15 min at 4 °C. Subsequently, samples were washed with FACS buffer, fixed, and permeabilized in CytoFix/CytoPerm Solution (BD Bioscience, 554714) for 15 min at 4 °C. Fixed cells were washed with 1x Perm/Wash Buffer (BD Bioscience, 554723), and stained for the following intracellular markers; anti-CD3-APC-Cy7 (clone UCHT1, Biolegend, 300426, diluted 1:40), anti-CD4-BV786 (clone SK3, BD Bioscience, 344642, diluted 1:40), anti-CD8-PE-

Cy7 (clone SK1, Biolegend, 344712, diluted 1:80), anti-IFNγ-PE (clone B27, Biolegend, 506507, diluted 1:40), and anti-TNFα-BV421 (clone Mab11, Biolegend, 502932, diluted 1:80). Each antibody was checked for performance and titrated before use. Following 15 min incubation at 4 °C, cells were washed thrice with PBS and acquired on a BD FACS Celesta with BD FACSDiva™ Software Version 8.0 (BD Bioscience). To minimize the signal from unspecific staining only T cells expressing IFNγ and TNFα were considered antigen-specific. The frequencies of antigen-specific T cells were calculated as negative-control-subtracted data. Possible longitudinal fluctuations in laser intensity were monitored daily before the experiment using fluorescent beads (Rainbow beads, Biolegend, 422905). If needed PMT voltages were adjusted to ensure constant signal intensity over time. The data were analyzed with the FlowJo Software version 10.0.7 (TreeStar). No technical replicates were performed due to the scarcity of the samples.

### Statistical Analysis

Statistical analysis was performed using GraphPad Prism software version 9.4.1. (681). Differences between the groups were assessed using the Kruskal-Wallis test with Dunn's correction for multiple testing. All tests were performed two-sided. Statistical significance is indicated by the following annotations: $*p < 0.05$, $**p < 0.01$, $***p < 0.001$, $****p < 0.0001$.

### Reporting summary

Further information on research design is available in the Nature Portfolio Reporting Summary linked to this article.

## Data availability

The data contain information that could compromise the privacy of research participants. Data sharing restrictions imposed by national and transnational data protection laws prohibit the general sharing of data. However, upon submission of a proposal to the corresponding author and approval of this proposal by (i) the principal investigator, (ii) the Ethics Committee of the University of Bonn, and (iii) the data protection officer of the University Hospital Bonn, data collected for the study can be made available to other researchers. A source data file containing the statistics presented in the figures and a Supplemental table containing demographic information are provided in this paper. Source data are provided with this paper.

## Code availability

No custom code or mathematical algorithm was generated for this study.

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

## Acknowledgements

The study was supported financially by the State of North Rhine-Westphalia, and the Viral NRW Network (grant number: CPS-1-1C acquired by

J.P. and H.S.). No other financial support was received. We thank the participants of this study who generously provided their samples.

## Author contributions

Conceptualization: J.P.; Methodology: J.P.; Investigation: J.P., J.Z., W.M.P., E.O. and K.P.; Resources: S.B. and W.M.P; Writing-original draft: J.P.; Writing-review & editing: J.P., H.S.; Funding acquisition: J.P., H.S.; Supervision: H.S.

## Funding

## Competing interests

The authors declare no competing interests. The idea, the plan, the concept, the protocol, the conduct, the data analysis, and the writing of the manuscript of this study were independent of any third parties, including the funding agency.
