## [Peer Review File · Nature Communications]

Vaccination impairs de novo immune response to omicron breakthrough infection, a precondition for the original antigenic sinReviewers' Comments:

Reviewer #1:

Remarks to the Author:

In the manuscript strong evidence is provided for original antigenic sin in SARS-CoV-2 infection, limiting induction of (neutralizing) antibody responses against novel epitopes upon omicron breakthrough infection. The authors convincingly demonstrate a reduced induction of binding as well as virus neutralizing antibodies and memory B cells towards novel epitopes in Wuhan strain vaccinated as well as infected persons, relatively to naive persons, upon subsequent omicron infection. T cell responses were not thus restricted and omicron specific peptides are recognized in all test groups. The experiments were well done and the results are robust and support the conclusions. While previous papers have also shown decreased omicron specific antibody binding and virus neutralization in previously Wuhan strain vaccinated and infected people the current paper provides an essential further step in supporting the concept of antigenic sin by: 1) studying besides antibody binding and neutralization also B memory cell responses and T cell responses, 2) performing competition ELISA and competition neutralization tests, 3) including a vaccine and infection naive group (omicron only) as well as omicron breakthrough Wuhan vaccinated and infected test groups.

Minor comments

1. Assessment of B cell memory percentages and T cell responses is performed for a relatively low number of people in the omicron only test group (6-7 out of 28 persons), while for the other test groups almost all selected persons were included. Was this small group of 6-7 people representative for the group as a whole? Were their age and gender distribution comparable?
2. Which S-protein (reagent specifics) was used as competitor in the competition ELISA?
3. In figure 2d representative plots are shown for number of plaques scored in the presence of increasing concentrations of inhibitory Wuhan proteins. What was the maximum number of plaques observed in cultures without any antibody? This could be indicated by a horizontal line in the graph and would show whether 100% competition can be achieved.
4. Are the data presented in figure 4b as total omicron specific response calculated by adding the response to the conserved peptides to the response against the mutated peptides?
5. In line 197 Fig. 4e should be Fig. 4i

Reviewer #2:

Remarks to the Author:

This is an interesting study that compared the vaccine responses in those vaccinated and infected vs those just vaccinated or just infected with SARS-CoV-2. Results from the study suggested that humoral responses to omicron breakthrough infections from those vaccinated were mainly recall of the vaccine-induced immune memory, both humoral and memory B cell responses against the altered regions of the omicron surface proteins were impaired. The T cell responses to mutated epitopes of the omicron spike protein were due to the high cross-reactivity of vaccine-induced T cells rather than the formation of a de novo response. The authors concluded that the imprinting of SARS-CoV-2 immunity by vaccination may lead to the development of original antigenic sin and impair the de novo immune responses to future variants.

There are a few issues in the manuscript, and suggestions for the authors to consider:

- The main text of the manuscript lacks the details of the participants characteristics, especially the details of the exposure history to SARS-CoV-2 and timing of the sample collection used for the subsequent analysis. Although the line listing is included in the supplementary materials, suggest the authors include a "summary table" in the main manuscript to describe the participant characteristics, including SARS-CoV-2 exposure history and the timing of sample collection of the 3 study groups.
- The main conclusions of this study are largely based on two in house developed methods: the in-house competitive ELISA and the in-house competitive plaque reduction neutralization assays. More qualification data should be included on these two methods, especially data demonstrating specificity

of the measurements. Suggest including qualification data at least in the supplementary materials, to demonstrate that these two methods can indeed differentiate omicron specific binding and neutralization responses rather than responses to the wild type SARS-CoV-2 (specificity).

- The current discussion section is largely a repeat of the results. Instead of repeating what have already been included in the results, in the discussion, the authors should expand on the implication of the study findings, strength and limitations of the study, and how the study findings can help refine future SARS-CoV-2 vaccine design and vaccination strategies, etc..

- From the results presented, the takeaway from the study is that the first exposure (initial imprinting by vaccination of the wild type strain vs initial imprinting by infection to variants) is important, furthermore, the sequence of the exposure (first infected or first vaccinated) could also be important in shaping the subsequent immune responses to SARS-CoV-2. However, this point is not very well articulated in the results and discussion. Suggest the authors re-designate the first study group (vaccinated then infected) as: "Vacc+O-inf" rather than "O-inf+vacc", because these participants were first imprinted by vaccination, then followed by infection, their immune responses could be different from those who are first imprinted by infection and then vaccination. Suggest expanding this point in the discussion.

- On this note, it would have been also helpful to include a study group that are first infected, and then vaccinated for comparison.

- The sample sizes of each study group are quite small O-inf+Vacc: n=37, Vacc: n=41 and O-Inf: n=28. It is also unclear why the sample sizes in each of the analysis are different, apparently not all samples were included in each analysis. Furthermore, the confidence intervals of the timing between the last exposure and sample collection are very wide in all experiments: Fig 1b, Fig 2b, Fig 3D, Fig 4D and 4H, these wide confidence intervals contributed to the "non-significant" difference when comparing the timing in these figures. these are limitations of the study should be stated.

Reviewer #1 (Remarks to the Author):

In the manuscript strong evidence is provided for original antigenic sin in SARS-CoV-2 infection, limiting induction of (neutralizing) antibody responses against novel epitopes upon omicron breakthrough infection. The authors convincingly demonstrate a reduced induction of binding as well as virus neutralizing antibodies and memory B cells towards novel epitopes in Wuhan strain vaccinated as well as infected persons, relatively to naive persons, upon subsequent omicron infection. T cell responses were not thus restricted and omicron specific peptides are recognized in all test groups. The experiments were well done and the results are robust and support the conclusions. While previous papers have also shown decreased omicron specific antibody binding and virus neutralization in previously Wuhan strain vaccinated and infected people the current paper provides an essential further step in supporting the concept of antigenic sin by: 1) studying besides antibody binding and neutralization also B memory cell responses and T cell responses, 2) performing competition ELISA and competition neutralization tests, 3) including a vaccine and infection naive group (omicron only) as well as omicron breakthrough Wuhan vaccinated and infected test groups.

Minor comments

Reviewer: Assessment of B cell memory percentages and T cell responses is performed for a relatively low number of people in the omicron only test group (6-7 out of 28 persons), while for the other test groups almost all selected persons were included. Was this small group of 6-7 people representative for the group as a whole? Were their age and gender distribution comparable?

Response: The reason for the low number of samples in the O-Inf group was that it was very difficult to get samples of individuals only infected with omicron SARS-CoV-2. By the time omicron emerged most of the people had already recovered from infection with previous variants or and received a vaccine. While we were able to get additional plasma samples from our cooperation partners, they did not collect the corresponding PBMC samples. In terms of gender (57% females and 43% males) and age (44±13) distributions the 7 individuals were representative of the whole O-Inf group. We now added this information to the manuscript section *Methods->Study cohort*.

Reviewer: Which S-protein (reagent specifics) was used as competitor in the competition ELISA?

Response: We thank the reviewer for bringing up this issue. We now added the missing reagent information to the manuscript section *Methods-> Measurement of plasma IgG specific for the omicron but not wild-type SARS-CoV-2 RBD*.

Reviewer: In Figure 2d representative plots are shown for number of plaques scored in the presence of increasing concentrations of inhibitory Wuhan proteins. What was the maximum number of plaques observed in cultures without any antibody? This could be indicated by a horizontal line in the graph and would show whether 100% competition can be achieved.

Response: We thank the reviewer for the suggestion. We now added the line representing the average of no-plasma controls to the corresponding graphs and modified the figure caption accordingly.

Reviewer: Are the data presented in figure 4b as total omicron specific response calculated by adding the response to the conserved peptides to the response against the mutated peptides?

Response: Yes, the total response to omicron spike peptides is calculated as a sum of responses to conserved and mutated peptide epitopes. We now made this clear in the manuscript section *Results-> The magnitude of T cell response to mutated regions of the omicron spike protein is comparable between the vaccinated individuals with or without omicron breakthrough and unvaccinated omicron-infected individuals.*

Reviewer: In line 197 Fig. 4e should be Fig. 4i

Response: We thank the reviewer for bringing up this mistake.

Reviewer #2 (Remarks to the Author):

This is an interesting study that compared the vaccine responses in those vaccinated and infected vs those just vaccinated or just infected with SARS-CoV-2. Results from the study suggested that humoral responses to omicron breakthrough infections from those vaccinated were mainly recall of the vaccine-induced immune memory, both humoral and memory B cell responses against the altered regions of the omicron surface proteins were impaired. The T cell responses to mutated epitopes of the omicron spike protein were due to the high cross-reactivity of vaccine-induced T cells rather than the formation of a de novo response. The authors concluded that the imprinting of SARS-CoV-2 immunity by vaccination may lead to the development of original antigenic sin and impair the de novo immune responses to future variants.

There are a few issues in the manuscript, and suggestions for the authors to consider:

Reviewer: The main text of the manuscript lacks the details of the participants characteristics, especially the details of the exposure history to SARS-CoV-2 and timing of the sample collection used for the subsequent analysis. Although the line listing is included in the supplementary materials, suggest the authors include a “summary table” in the main manuscript to describe the participant characteristics, including SARS-CoV-2 exposure history and the timing of sample collection of the 3 study groups.

Response: We agree with the reviewer and we have now incorporated a graphical abstract summarizing the time points of sample collections and antigen exposure into Figure 1. Additionally, graphs comparing the time passed between the last exposure and sample collection between the groups are included in every figure. We summarize the gender and age compositions of each group in the *Methods section-> Study cohort* since the inclusion of small tables is discouraged by the journal’s formatting guidelines “Unnecessary figures should be avoided: data presented in small tables or histograms, for instance, can generally be stated briefly in the text instead“. Detailed demographic information is provided in Supplemental Table 1.

Reviewer: The main conclusions of this study are largely based on two in house developed methods: the in-house competitive ELISA and the in-house competitive plaque reduction neutralization assays. More qualification data should be included on these two methods, especially data demonstrating specificity of the measurements. Suggest including qualification data at least in the supplementary materials, to demonstrate that these two methods can indeed differentiate omicron specific binding and neutralization responses rather than responses to the wild type SARS-CoV-2 (specificity).

Response: We agree with the reviewer and now included the validation data for both assays as supplemental material. Please see the attached file *Validation_competitive assays*.

Reviewer: The current discussion section is largely a repeat of the results. Instead of repeating what have already been included in the results, in the discussion, the authors should expand on the implication of the study findings, strength and limitations of the study, and how the study findings can help refine future SARS-CpV-2 vaccine design and vaccination strategies, etc.

Response: It is common practice to repeat the main findings at the beginning of a discussion and then put them in context with other findings. However, we agree with the reviewer and now expanded the discussion by adding strengths and limitations and possible implementations of the results.

Reviewer: From the results presented, the takeaway from the study is that the first exposure (initial imprinting by vaccination of the wild-type strain vs initial imprinting by infection to variants) is important, furthermore, the sequence of the exposure (first infected or first vaccinated) could also be important in shaping the subsequent immune responses to SARS-CoV-2. However, this point is not very well articulated in the results and discussion. Suggest the authors re-designate the first study group (vaccinated then infected) as: "Vacc+O-inf" rather than "O-inf+vacc", because these participants were first imprinted by vaccination, then followed by infection, their immune responses could be different from those who are first imprinted by infection and then vaccination. Suggest expanding this point in the discussion.

Response: We agree with the reviewer. This study investigates how the previous vaccination with the WT-based vaccine affects the immune response to subsequent infection with the omicron variant and not the other way around. We now renamed the group and made this clear throughout the manuscript. We also added a graphical representation of the study design including exposure history to Figure 1.

Reviewer: On this note, it would have been also helpful to include a study group that are first infected, and then vaccinated for comparison.

Response: Unfortunately we did not collect samples of individuals that would be first infected with omicron and then vaccinated as this was not the core question of this study (does repeated vaccination with the WT-based vaccine imprint the response to infection with omicron variant that has a heavily mutated spike protein). We now made this clear in the Introduction. In addition, it will be hardly possible to find such subjects as vaccination campaigns were already long underway and we do not have samples of a single case, who was first infected with omicron and later decided to get vaccinated. Most of the individuals who did not get vaccinated until omicron emergence were against getting vaccinated and would most probably not get their first vaccine dose after the infection. Even if they would do so, by that

time WT-based vaccines were mostly replaced by omicron-adapted vaccines. Overall, we believe that there are very few individuals who recovered from omicron infection and subsequently received a WT-based vaccine, therefore, studying immune imprinting in this context would be difficult and not reflect the real-world situation.

Reviewer: The sample sizes of each study group are quite small O-inf+Vacc: n=37, Vacc: n=41 and O-Inf: n=28. It is also unclear why the sample sizes in each of the analysis are different, apparently not all samples were included in each analysis. Furthermore, the confidence intervals of the timing between the last exposure and sample collection are very wide in all experiments: Fig 1b, Fig 2b, Fig 3D, Fig 4D and 4H, these wide confidence intervals contributed to the “non-significant” difference when comparing the timing in these figures. these are limitations of the study should be stated.

Response: The reason for the low number of samples was that it was very difficult to get samples of individuals only infected with omicron SARS-CoV-2 (O-Inf group). By the time omicron emerged most of the people had already recovered from infection with previous variants or/and received a vaccine. While we were able to get additional plasma samples from our cooperation partners, they did not collect the corresponding PBMC samples so we have variable sample sizes for different assays. We now added this as a limitation of the study. Another factor contributing to the difference in sample sizes is that samples that were negative for omicron-specific antibodies/B cells/T cells could not be considered for the assays measuring the proportion of the omicron- but not WT-specific antibodies/B cells/T cells. In some instances, samples were not measured due to insufficient sample material. The whiskers in plots Fig 1b, Fig 2b, Fig 3D, Fig 4D and 4H represented range not confidence intervals. We now change these graphs to comply with the editorial policy. We agree with the reviewer that the wide range could contribute to the lack of significance and added this to the discussion.

Reviewers' Comments:

Reviewer #1:

Remarks to the Author:

All questions were properly addressed. There are no further comments.

Reviewer #2:

Remarks to the Author:

The authors have addressed the reviewer questions in the revision.